# EggBlock: Design and Implementation of Solar Energy Generation and Trading Platform in Edge-Based IoT Systems with Blockchain

**DOI:** 10.3390/s22062410

**Published:** 2022-03-21

**Authors:** Subin Kwak, Joohyung Lee, Jangkyum Kim, Hyeontaek Oh

**Affiliations:** 1Samsung Electronics, Yeongtong-Gu, Suwon 10285, Korea; starfishda54@gmail.com; 2School of Computing, Gachon University, Seongnam 13120, Korea; 3School of Electrical Engineering, KAIST, Daejeon 34141, Korea; 4Institute for Information Technology Convergence, KAIST, Daejeon 34141, Korea; hyeontaek@kaist.ac.kr

**Keywords:** solar energy generation, energy trading, auction theory, testbed, measurement study, Internet of Things, blockchain, reinforcement learning

## Abstract

In this paper, to balance power supplement from the solar energy’s intermittent and unpredictable generation, we design a solar energy generation and trading platform (EggBlock) using Internet of Things (IoT) systems and blockchain technique. Without a centralized broker, the proposed EggBlock platform can promote energy trading between users equipped with solar panels, and balance demand and generation. By applying the second price sealed-bid auction, which is one of the suitable pricing mechanisms in the blockchain technique, it is possible to derive truthful bidding of market participants according to their utility function and induce the proceed transaction. Furthermore, for efficient generation of solar energy, EggBlock proposes a Q-learning-based dynamic panel control mechanism. Specifically, we set the instantaneous direction of the solar panel and the amount of power generation as the state and reward, respectively. The angle of the panel to be moved becomes an action at the next time step. Then, we continuously update the Q-table using transfer learning, which can cope with recent changes in the surrounding environment or weather. We implement the proposed EggBlock platform using Ethereum’s smart contract for reliable transactions. At the end of the paper, measurement-based experiments show that the proposed EggBlock achieves reliable and transparent energy trading on the blockchain and converges to the optimal direction with short iterations. Finally, the results of the study show that an average energy generation gain of 35% is obtained.

## 1. Introduction

The use of solar energy is considered as a promising renewable energy source. Solar energy has various advantages in terms of increasing energy efficiency and reducing greenhouse gas emissions [1]. In the conventional centralized power grid system, several problems can occur, such as energy loss owing to the transmission process and power instability due to the peak power demand [2]. Using solar energy as a distributed energy resource, it is possible to minimize the transmission loss and supply energy to the consumer more efficiently. However, the solar panel has a problem in that it is sensitive to changes in the surrounding environment, and it is difficult to arbitrarily control the amount of energy generation [3,4].

To address these challenges, various studies have been conducted in the literature, aiming for efficient energy trading mechanisms [5,6,7]. Specifically, using a centralized system manager, market-based energy trading models have been suggested for balancing demand and generation. However, such interventions of the centralized system cause additional participation fees and security issues in transaction records. Thus, to alleviate these problems, blockchain technology-based distributed energy trading models have been proposed as one of the promising technologies in the smart grid system [8,9,10]. Nevertheless, research on implementation-based energy trading in distributed solar energy generation is limited. Most of the studies were confined to numerical or simulation-based studies. As a result, no conventional studies have addressed practical designs, such as the implementation of solar energy generation and trading platforms that consider AI-based automatic energy generation and blockchain-based secure energy trading at the same time. Therefore, we deal with an operation method of solar energy as one of promising renewable energy source, considering the actual environment and the residual energy transaction platform using the actual dataset.

In this study, we design a solar energy generation and trading platform in Internet of Things (IoT) systems using blockchain (EggBlock), which promotes distributed energy trading without introducing a centralized broker and supports efficient energy generation. The contributions of this study are summarized as follows:To support reliable energy trading among users without the participation of a centralized broker, the EggBlock platform using Ethereum for blockchain is proposed in this paper. In order to determine a reasonable transaction of the generated renewable energy, we infer the determination of transaction price and the amount of energy in the market according to the second price sealed-bid auction mechanism. Furthermore, for efficient generation of solar energy, the EggBlock proposes the Q-learning based dynamic panel control mechanism. Specifically, we consider a model-free-based Q-learning algorithm, where the state and the action are the current position of the solar panel and the angle at which the solar panel will move to the next time step, respectively. Finally, the reward is designed as the amount of solar power that the solar panels can obtain under the given state.We implement the proposed EggBlock platform using Ethereum for blockchain, which enables Android smartphones to monitor contracts of energy trading in a real-time manner. Furthermore, we build an IoT hardware testbed equipped with a solar panel to generate and deliver energy for trading.The measurement-based experiments in the testbed show that the proposed EggBlock achieves reliable and transparent energy trading using blockchain, and it converges to the optimal direction with short-iterations. Finally, the results of the study show that an average energy generation gain of 35% is obtained.

The remainder of this study is organized as follows. We review existing research in Section 2. After the introduction of the overall system model in Section 3, the Q-learing algorithm is introduced in detail in Section 4. Section 5 contains a detailed description of the actual implementation, while Section 6 contains a detailed description of the experimental results. Finally, Section 7 concludes with a detailed description of our system’s usage area and future plans.

## 2. Related Works

Currently, renewable energy accounts for a large portion of the total energy generation [11]. It is advantageous in terms of environmental sustainability, low maintenance cost, and ease of installation in urban areas [12]. However, the energy generation of renewable energy is unstable due to various uncertain factors, such as weather, cloud movement, or solar irradiation [13]. Therefore, various studies have been conducted to manage unstable energy generation with the installation of additional facilities, such as energy storage systems or auxiliary generators. However, the construction of such additional facilities requires considerable cost and time [14]. Thus, various energy trading approaches in smart grid systems have been proposed for reliable and efficient energy use by balancing demand and generation [5,6,7]. The work in [5] provided a game-theoretical analysis for a distributed energy trading mechanism in which multiple microgrids re-sell or store surplus energy to maximize their own profit. By representing the Nash equilibrium of all players, the authors analyzed individual players’ strategies in each market environment. In [6], Zhang et al. proposed a peer-to-peer (P2P) energy trading model to improve the local energy balance. Here, the authors considered a practical low-level voltage microgrid environment to analyze its effectiveness. In [7], Pei et al. introduced the concept of a two-stage market model to maintain the balance of energy supply and demand in the entire system and region. Specifically, using a Monte Carlo sampling method, the authors alleviated the effect of the uncertainties of renewable energy. Additionally, by showing the various simulation results, the authors verified that the proposed market framework applies to an actual system.

Nevertheless, in most of the previous energy trading studies, interventions of the system manager should manage such an energy trading market. However, such interventions also resulted in additional participation fees and security issues of transaction records. Therefore, blockchain technology-based distributed energy trading model without a centralized manager have been proposed as solutions to alleviate such issues in the smart grid system [8,9,10,15,16,17]. In [8], Mihaylov et al. proposed an energy trading method for energy prosumers and consumers using the proposed NRGcoin. The authors argued that the introduction of NRGcoin can solve the security issues associated with energy trading and the fiat money conversion problem. In [9], Mengelkamp et al. developed energy transaction models and demonstrated the simulation environment in the region using the private blockchain method. The authors show the market operation results by analyzing the relationship between the proportion of solar energy generation in the total energy supplement and market price from an economic perspective. On the other hand, with the implementation of an actual experimental system, Zhang et al. proposed a real-time energy transaction system by incorporating the concepts of prioritization and cryptocurrency [10]. By presenting the numerical results, the authors showed that the proposed system can be beneficial to market participants from both monetary and non-monetary perspectives. In addition, unlike most conventional studies that use cryptocurrency as an alternative currency, the proposed cryptocurrency in the paper has the abilities to convert fiat currency and change the physical power flow. Vehicle to grid (V2G) network is one of the environments in smartgrid system that is appropriate to use blockchain technology in, as depicted in [15]. Hassija et al. addressed the direct acyclic graph-based V2G network (DV2G), which is organized with lightweight blockchain protocol. Showing the mechanism and numerical results, authors proved that the proposed method is highly scalable and supports the micro-transaction that is required in V2G network. By introducing the blockchain technique, it is possible to operate the system according to the consensus of smart contract between the participants without the intervention of a third party. In [16], Xi Chen et al. proposed energy transaction among the renewable energy and electric vehicle to minimize the burden of the power system operator. Through the blockchain-based EV incentive system, authors addressed that it is possible to maximize the utilization of renewable energy and manage the power system more effectively. A similar approach to this paper is dealt in [17]. In the paper, Hassan et al. proposed auction and blockchain-based energy transaction scheme to maximize the energy producer’s revenue in the system. Here, the authors insist that it is possible to provide moderate cost, secure, and private auction schemes for microgrids using the blockchain technology. Applying the advantages of blockchain to maximize the profit of various market participants and increase the stability of energy transaction is similar to this paper. However, our paper has the distinction of maximizing the amount of power generation by controlling the angle of the solar panel and considering the actual electricity tariff to facilitate application in the real environment.

## 3. Proposed System

As illustrated in Figure 1, the proposed EggBlock system model considers that there are multiple edge-based IoT devices as users who participate in energy trading using the generated energy. We set the environment in which the edge-based IoT devices are equipped with solar panels for energy generation and energy storage to store redundant energy for future use. Therefore, the proposed EggBlock system consists of two parts: (1) energy generation and (2) energy trading.

In the energy generation part, we deal with a method that can maximize energy generation by controlling the angle of the solar panel. Here, the intensity of light over various locations is measured to determine the direction of the solar panel where a BH1750 photo resistor controlled by Raspberry Pi is used. Such measured data are used for the Q-learning algorithm, which maximizes the expected reward (i.e., energy generation) by determining the optimal solar panel angle under the given state space.

In the energy trading part, we propose a blockchain-based decentralized energy trading model that is organized without a centralized broker. In this study, we design a transaction environment using an Ethereum-based decentralized application (Dapp) platform. Here, the use of Dapp is suitable for the proposed energy transaction model because it has various advantages (e.g., zero downtime, privacy, resistance to censorship, complete data integrity, and trustless computation/verifiable behavior) [18,19]. Based on these advantages, participants in the market cannot cheat or falsify transactions and have a reliable transaction service.

### 3.1. Energy Generation

To maximize energy generation by adjusting the position of the solar panels, we use a reinforcement learning (RL) framework. Here, we use two Raspberry Pi boards as edge nodes to control the solar panel: (1) collecting solar energy data and (2) using the collected data to train Q-Learning and adjusting the position of solar panels through stepper motors.

In the first edge node, solar energy data collection is performed using a photo register, which can be used for training the RL model. The collected data are sent on every time step to the second edge node.

To use the data collected from the first edge node, a Q-table is defined using a model-free RL framework at the second edge node [20]. For every time step, the second edge node selects an optimal action by referring to the Q-table and transmits the action value to a stepper motor to efficiently control the position of the solar panel. The Q-table is updated at every time step based on the received reward corresponding to the state so that it converges to the optimal action corresponding to the state. Here, when the stepper motor moves the solar panel based on the received action value, the reward occurs according to the action process and updates the Q-table. The detailed process of the RL framework will be covered in Section 4.

### 3.2. Energy Trading

The energy obtained from solar panel is sold to bidders at a price determined based on the auction mechanism. For reliable energy transactions between sellers and buyers, the energy trading platform is implemented through a blockchain using Ethereum. According to the Ethereum-based smart contract, transactions proceed among sellers and buyers with their purpose and status. Accordingly, if the transaction is conducted via the smart contract, the blockchain guarantees the authenticity and the reliability of the transaction execution without requiring a centralized broker [21].

#### 3.2.1. Determination of Transaction Based on Sealed-Bid Auction Mechanism

The energy generated by the photovoltaic (PV) generator is sold at an appropriate price to buyers who need it. At this time, in order to determine the transaction price, a second price sealed-bid auction mechanism is proposed in this paper. There are two reasons for choosing the second price sealed-bid auction mechanism in this paper. First, it is possible to minimize the exposure of information about the buyer’s bidding strategy (e.g., bidding price, current status, policy, etc.). Second, buyers adopt a truthful bidding strategy in which their expected price is equal to their bidding price. Since it can be seen that the advantages of such mechanism are efficient in the blockchain-based energy transaction model, we use the method and predict the bidding price of buyers.

In Figure 2, we represent the overall structure of auction-based market model. Since the proposed market is operated separately from the conventional energy market, it is necessary to consider the market data in actual system. In the figure, we could check that there are two different participants in the market. The energy producer is a seller who has renewable energy with a storage device to sell residual power. The *energy consumer* is a buyer who wants to purchase energy from the producer to minimize the overall cost. Here, the consumer might be a prosumer with solar energy or simple electricity consumer who wants to purchase energy in a cost-effective manner through EggBlock rather than purchasing from the conventional market. As depicted in the above statement, a transaction of the proposed scheme is the progressed auction mechanism. Therefore, the auction is began through the disclosure of the amount of energy sold by the seller.

In this paper, we use a second price sealed-bid auction mechanism that the seller sells the energy to the buyer with the highest bidding price for the second highest price. The characteristics of the auction called sealed-bid makes the buyer decide the bidding price by only considering it’s own profit. In addition, considering the bidding mechanism, it can be seen that the buyer makes a truthful bidding in the proposed system [22,23]. Therefore, the details of the utility function to determine each participant’s strategy are depicted as follow:

**Seller:** In the case of a seller, it is possible to determine the amount and price of energy to be sold considering the current profit and the future value achieved by storing the energy. Here, the utility function of the seller could be depicted as: (1)$$\begin{array}{ccc} \max_{p} & & \hat{U}\left(p\right)=C_{sell}\hat{E}p+C_{op}ln\left(1+\hat{E}\eta \left(1-p\right)\right] \end{array}$$(2)$$\begin{array}{ccc} s.t. & & 0\leq p\leq 1. \end{array}$$

In Equation (Equation 1), the first term $C_{sell}\hat{E}p$ refers to the profit that the seller achieves through selling energy. Here, $C_{sell}$ refers to the unit price for selling energy, and $\hat{E}$ is the energy generated from the solar panel. In addition, *p* is the decision variable of the seller that means the portion of energy to sell in the market. Furthermore, the second represents the satisfaction the seller attained by storing the energy $\hat{E}\left(1-p\right)$ in the storage system [24]. Here, a weight factor $C_{op}$ is added to transform the seller’s satisfaction level into the monetary aspect. Since the formula is a concave model consisting of one decision variable, the optimal value can be calculated as follows:

**Derivation:** To obtain an optimal solution of the problem, we differentiate Equation (Equation 1) by *p* as:(3)$$\begin{array}{cc} \frac{\partial \hat{U}\left(p\right)}{\partial p} & =\alpha \hat{E}-\frac{\hat{E}\eta }{1+\hat{E}\eta \left(1-p\right)}, \end{array}$$
(4)$$\begin{array}{cc} \frac{\partial^{2}\hat{U}\left(p\right)}{\partial p^{2}} & =-\frac{\hat{E}\eta^{2}}{{\left(1+\hat{E}\eta \left(1-p\right)\right)}^{2}}. \end{array}$$

To organize the formula more easily, we transform the $\frac{C_{sell}}{C_{op}}$ into the auxiliary variable α. Assuming that the value of $C_{op}$ is larger than $C_{sell}$, we could set 0≤α≤1 in the equation. In addition, since the auxiliary variable η is positive, the second derivation of utility function in Equation (Equation 4) becomes a negative value. Since the utility function of the seller is organized with a single decision variable, it is possible to prove that the utility function of the seller is concave. Therefore, the optimal value $p^{*}$ is calculated as 0 according to the first derivation in Equation (Equation 3). In this way, we could calculate the optimal value as follows:(5)$$\begin{array}{c} p^{*}=\max\left(\min\left(1-\frac{1}{\hat{E}}\left(\frac{\left(\eta -\alpha \right)}{\alpha \eta }\right),1\right),0\right). \end{array}$$

**Buyer:** In the case of a buyer, the main purpose is minimizing the overall cost by submitting the proper bidding price to the auction market.
(6)$$\begin{array}{cc} U_{i}\left(b_{i}\right) & =\left(1-e^{-\gamma_{i}b_{i}}\right)\left(\left(p_{s}-b_{i}\right)\min\left(D_{i},E\right)-b_{i}\max\left(E-D_{i},0\right)\right)-e^{-\gamma_{i}b_{i}}p_{s}D_{i}. \end{array}$$

In Equation (Equation 6), $\left(1-e^{\gamma_{i}b_{i}}\right)\left(\left(p_{s}-b_{i}\right)\min\left(D_{i},E\right)-b_{i}\max\left(E-D_{i},0\right)\right)$ refers to the profit that a buyer *i* can achieve from the auction market. In the equation, $\left(1-e^{\gamma_{i}b_{i}}\right)$ refers to the probability that the buyer *i* could achieve the energy from the auction progress. Here, $\gamma_{i}$ is an auxiliary variable that determines the probability of successful bidding according to the bidding price submitted by the buyer *i*. In addition, $\left(p_{s}-b_{i}\right)\min\left(D_{i},E\right)$ is the profit that the buyer *i* achieved from the auction process. For the case that the buyer *i* successes the bidding process, the buyer could achieve the monetary profit $\left(p_{s}-b_{i}\right)\min\left(D_{i},E\right)$ and get the loss $b_{i}\max\left(E-D_{i},0\right)$ when the purchasing energy *E* is higher than required energy $D_{i}$. Furthermore, the last term $-e^{\gamma_{i}b_{i}}p_{s}D_{i}$ is the cost that the buyer *i* has to pay when the buyer fails in the auction.

*Case 1:* For the case that a buyer *i*’s required energy $D_{i}$ is larger than purchasing energy *E*, we could set the utility function of buyer *i* as follows:(7)$$\begin{array}{cc} U_{i}\left(b_{i}\right) & =\left(1-e^{-\gamma_{i}b_{i}}\right)\left((p_{s}-b_{i})E\right)-e^{-\gamma_{i}b_{i}}p_{s}D_{i}. \end{array}$$

According to the second derivative result of the buyer’s utility function, it is possible to check that the utility function is organized with the concave function. To find the optimal $b_{i}^{*}$ that maximizes the profit of buyer *i*’s profit, we could differentiate the Equation (11) as:(8)$$\begin{array}{cc} \frac{\partial U(b_{i})}{\partial b_{i}} & =\gamma_{i}e^{-\gamma_{i}b_{i}}\left(p_{s}E-b_{i}E\right)-\left(1-e^{-\gamma_{i}b_{i}}\right)E+e^{-\gamma_{i}b_{i}}\gamma_{i}p_{s}D_{i}=0. \end{array}$$

Here, we could reorganize the formula as:(9)$$\begin{array}{cc} e^{\gamma_{i}b_{i}} & =1+\gamma_{i}\left(p_{s}-b_{i}\right)+\frac{1}{E}\left(\gamma_{i}p_{s}D_{i}\right). \end{array}$$

According to the Taylor series equation, we could change $e^{\gamma_{i}b_{i}}=1+\gamma_{i}b_{i}+\frac{\gamma_{i}^{2}b_{i}^{2}}{2!}+\frac{\gamma_{i}^{3}b_{i}^{3}}{3!}+\ldots$. Here, we assume that values after $\frac{\gamma_{i}^{3}b_{i}^{3}}{3!}$ are too small to be 0. Therefore, we could find the optimal value through the following equation.
(10)$$\begin{array}{c} 1+\gamma_{i}b_{i}+\frac{\gamma_{i}^{2}b_{i}^{2}}{2}=1+\gamma_{i}\left(p_{s}-b_{i}\right)+\frac{1}{E}\left(\gamma_{i}p_{s}D_{i}\right)\\ {\left(b_{i}+\frac{2}{\gamma_{i}}\right)}^{2}-\frac{4}{\gamma_{i}^{2}}-\frac{2p_{s}}{\gamma_{i}}-\frac{2}{E\gamma_{i}}p_{s}D_{i}=0\\ ∴b_{i}^{*}=\sqrt{\frac{4}{\gamma_{i}^{2}}+\frac{2p_{s}}{\gamma_{i}}+\frac{2}{E\gamma_{i}}p_{s}D_{i}}-\frac{2}{\gamma_{i}} \end{array}$$

*Case 2:* For the case that purchasing energy *E* is larger than the buyer *i*’s required energy $D_{i}$, we could set the utility function as follows:(11)$$\begin{array}{cc} U_{i}\left(b_{i}\right) & =\left(1-e^{-\gamma_{i}b_{i}}\right)\left((p_{s}D_{i}-b_{i}E\right)-e^{-\gamma_{i}b_{i}}p_{s}D_{i}. \end{array}$$

By differentiating the utility function, we could get the optimal bidding price as:(12)$$\begin{array}{c} \gamma_{i}e^{-\gamma_{i}b_{i}}\left(p_{s}D_{i}-b_{i}E\right)-\left(1-e^{-\gamma_{i}b_{i}}\right)E+\gamma_{i}e^{-\gamma_{i}b_{i}}p_{s}D_{i}=0\\ \gamma_{i}e^{-\gamma_{i}b_{i}}\left(2p_{s}D_{i}-b_{i}E+\frac{E}{\gamma_{i}}\right)-E=0\\ e^{\gamma_{i}b_{i}}=\frac{2p_{s}D_{i}\gamma_{i}}{E}-\gamma_{i}b_{i}+1 \end{array}$$(13)$$\begin{array}{c} 1+\gamma_{i}b_{i}+\frac{\gamma_{i}^{2}b_{i}^{2}}{2}=\frac{2p_{s}D_{i}\gamma_{i}}{E}-\gamma_{i}b_{i}+1\\ b_{i}^{2}+\frac{4b_{i}}{\gamma_{i}}-\frac{4p_{s}D_{i}}{E\gamma_{i}}=0\\ {\left(b_{i}+\frac{2}{\gamma_{i}}\right)}^{2}-\frac{4}{\gamma_{i}^{2}}-\frac{4p_{s}D_{i}}{E\gamma_{i}}=0\\ ∴b_{i}^{*}=\sqrt{\frac{4}{\gamma_{i}^{2}}+\frac{4p_{s}D_{i}}{E\gamma_{i}}}-\frac{2}{\gamma_{i}} \end{array}$$

In this way, it is possible to predict the strategies of seller and buyers in the market. Since we use the second price sealed-bid auction, each participant conducts a truthful bidding according to the optimal value of its utility function. After participants decide their strategy, the amount of transaction energy set by the seller is sold to the person who bids the highest price by paying the second highest bidding price. Here, information about the whole of the participants’ strategy and transaction results are stored in the block.

#### 3.2.2. Architecture of Smart Contract

Ethereum is a global, decentralized open-source blockchain featuring smart contract functionality, where ether (ETH) in Ethereum is the cryptocurrency generated by Ethereum miners as a reward for computations for adding blocks to the blockchain. The detailed Ethereum architecture is illustrated in Figure 3. The term user in Figure 3 represents both buyers and sellers who participate in a transaction. Correspondingly, when a user transacts via Ethereum, the user must first connect to a node and refer to the Ethereum client through a web browser powered by MetaMask [25].

The Ethereum clients, who have a distributed database and are interconnected to the Ethereum Network, receive information when the block is created owing to a new transaction. In other words, Ethereum clients are nodes of a blockchain network, called blockchain nodes, which allow general users to connect to the blockchain. Users will be able to obtain blockchain information or use smart contracts through the Ethereum client. The smart contract is a script that implements the contents of a contract using the code and allows the contract to be automatically fulfilled when the conditions are met. The smart contract created by one user is stored in blocks through the Ethereum client. Therefore, all nodes in the blockchain network have the same smart contract code. The process of creating smart contracts includes the following:Smart contract coding: Code the contents of the contract you want to include in the smart contract.Connect with Ethereum client: The written code is placed on the Ethereum virtual machine (EVM) of the Ethereum client.Compile the implemented code: EVM byte code will provide the compilation result.Smart contract distribution: Add the compiled EVM byte code to the block as a transaction and register to the blockchain.

In this process, when the smart contract is registered to the blockchain, all Ethereum clients on the blockchain will have the byte code of the smart contract. Then, the Ethereum client will be able to run a registered smart contract on its EVM. Additionally, if a transaction occurs, the content of a smart contract will change. Thus, after the transaction, other nodes can access the smart contract so that they can change the content.

#### 3.2.3. Transaction Process

Figure 4 shows the sequence diagram for the energy trading. In the energy trading platform, some participants have superfluous energy that they wish to sell to the platform as sellers, whereas others do not have sufficient energy to meet their demands and must buy the shortfall from the platform as buyers. Specifically, first, the buyer defines and sends Ether and code so that smart contracts can be created.

A smart contract is run on top of the EVM. The terms of the transaction stored in the smart contract are disclosed to everyone who can participate in the transaction. For reliable transactions, sellers should prove the amount of energy they have. The amount of proven energy is returned to the token, which is held by the seller. At this time, the token is used only to prove the amount of energy without monetary value. If there is a seller who has sufficient energy to sell as much as the buyer needs, the seller sends a token with energy to the buyer. After the buyer receives the energy as required, it sends the completed message receipt to the smart contract account. Finally, the smart contract delivers the ETHs stored in the transactions to the seller. Simultaneously, the token sent to the purchaser is extinguished.

Through this process, two participants can make a safe transaction without third-party intervention, and this process is transparently recorded in the blockchain.

## 4. Model-Free Q-Learning

Model-free RL aims to achieve efficient control of the solar panel’s movement so that the solar panel can collect the maximum energy when compensation is unknown. To design the model-free RL algorithm, a Markov decision process (MDP) framework is considered, which requires a description of states, actions, and rewards [20]. We consider the discrete state and action spaces where the state and action refer to the current location of the solar panel and the amount of angle to be moved. Here, the goal of the agent (i.e., solar panel) is to learn the policy π, which allows the agent to choose actions that can maximize long-term cumulative rewards (i.e., energy generation) where states are given.

### 4.1. Markov Decision Process

The environment for the MDP consists of an agent, a BH1750 photo register, and a solar energy battery capable of storing energy. In this study, we assume an environment in which time is divided into consecutive fixed-length intervals. In each time interval, the agent, which is a solar panel, determines the action. Here, we define a set of time steps that identifies the intervals as T={0,1,2,...}. Then, the agent determines the action at the next time step to maximize the expected cumulative rewards based on photo resistor observation and previous experience. Specifically, the angle movement of the solar panel is controlled through the stepper motor. Then, the chosen action is transmitted to the stepper motor to control the angle of the solar panel. A detailed description of the states, actions, and rewards for the proposed MDP framework is provided as follows.

**State:** State means the current location of the agent. The location of the agent in the *t* time step is received as a scalar value of $s_{t}$. The set of states is denoted by *S* and marked as $S=\{s_{1},s_{2},s_{3},\ldots ,s_{t}\}$. We denote the state of the system in time step *t* as $s_{t}$, which is given as follows: (14)$$\begin{array}{c} s_{t}=\left[\mathrm{solar}{\mathrm{panel}}^{\prime }s\mathrm{angle}\mathrm{in}\mathrm{time}\mathrm{step}t\right] \end{array}$$

**Action:** Action means the angle at which the motor must move at $s_{t}$. The action value in the time step *t* can be expressed as a scalar value $a_{t}$. The set of actions is named *A* and marked as $A=\{a_{1},a_{2},a_{3},\ldots ,a_{t}\}$. Consider the state of the system in time step *t* denoted by $a_{t}$, which is given as follows:(15)$$\begin{array}{c} a_{t}=\left[\mathrm{angle}\mathrm{to}\mathrm{move}\mathrm{in}\mathrm{time}\mathrm{step}t\right] \end{array}$$

**Reward:** Reward $R(s_{t},a_{t})$ obtained by taking action $a_{t}$ at state $s_{t}$ is the total amount of energy that can be obtained from state $s_{t}$. Our goal is to maximize the total energy generation, which can be expressed in a formula to maximize the expected rewards of the solar panel.
(16)$$\begin{array}{c} R\left(s_{t},a_{t}\right)=\left[\mathrm{amount}\mathrm{of}\mathrm{energy}\mathrm{can}\mathrm{be}\mathrm{obtained}\mathrm{in}\mathrm{time}\mathrm{step}t\right]. \end{array}$$

### 4.2. Q-Learning

The state and action pairs (s,a) applied in this study are mapped through policy π derived using Q-learning. $Q^{\pi }\left(s,a\right)$ is the cumulative reward for taking action *a* from a state *s* and follows policy π accordingly. $Q^{\pi }$ is specified as follows:(17)$$\begin{array}{c} Q^{\pi }\left(s,a\right)=E\left[\sum_{t=0}^{\infty }\gamma_{i}^{t}R\left(s_{t},\pi \left(s_{t}\right)\right)|s_{0}=s,a_{0}=a\right]. \end{array}$$
where $\gamma_{i}\in \left(0,1\right)$, which is a hyper-parameter of the Q-function, is the discount factor. Therefore, maximizing the cumulative reward is the same as finding a policy that maximizes the Q-function. The optimized Q-function is called $Q^{*}$, which satisfies the Bellman equation.
(18)$$\begin{array}{c} Q^{*}\left(s,a\right)=R\left(s,a\right)+\gamma_{i}\cdot E_{s^{\prime }}\left[\max_{a}Q^{*}\left(s^{\prime },a\right)\right]. \end{array}$$

The purpose of the Q-learning algorithm is to learn the optimal $Q^{*}$ in the observation sequence $\left(s_{t},a_{t},R_{t+1},s_{t+1}\right)$. The optimal policy $\pi^{*}$ can be computed using $Q^{*}$ as follows:(19)$$\begin{array}{c} \pi^{*}\left(s\right)=\underset{a}{arg max}Q^{*}\left(s,a\right). \end{array}$$

The Q-learning algorithm was implemented as follows. At each time step *t*, the agent updates the Q-function $Q_{t}$ as follows:(20)$$Q\left(s_{t},a_{t}\right)=\left(1-\alpha_{t}\right)Q\left(s_{t},a_{t}\right)+\alpha_{t}\left(R_{t}+\gamma_{i}\max_{a}Q\left(s_{t+1},a_{t}\right)\right),$$
where $\alpha_{t}$ denotes the learning rate. If state-action pairs are sampled infinitely and under suitable conditions on the learning rate, $Q_{t}$ will converge to the optimal Q-function *Q* [26].

As described in Algorithm 1, we set the parameter *K* as the time range representing the daily energy collection time. Furthermore, ϵ is the hyper-parameter value of ϵ-greedy, which indicates the probability that a solar panel randomly selects an action. As described in the pseudocode, each episode consists of a one-day energy generation. *t* is the time interval for calculating the reward. Therefore, in each time step *t*, the solar panel obtains the current state and selects the action. There are two choices when selecting an action. Exploring the policy may speed up training, but it may cause problems of falling into a local optimum due to the inability to explore new routes. Therefore, we apply an e-greedy method that can select a random action with a certain probability. After selecting the action, the reward is calculated according to the state and action. Subsequently, the Q-table is updated according to Equation (Equation 20). Then, we go to the next time step and iterate this process.
**Algorithm 1 **Q-Learning based solar panel control method.1:Setting *K* range2:Initial $\gamma_{i}$, ϵ3:Initial action-value function $Q_{0}$ and *t*4:**for** each episode **do**5:    **while** t≤K **do**6:        Get current state $s_{t}$7:        Select action
$$a_{t}=\left\{\begin{array}{cc} \mathrm{random}\;a, & \mathrm{if}\;\mathrm{probability}\;\epsilon ,\\ {arg max}_{a}Q\left(s_{t},a_{t}\right), & \mathrm{else}, \end{array}\right.$$8:        Execute action $a_{t}$9:        Calculate reward $R(s_{t},a_{t})$10:        State $s_{t}$ transfer to the next state $s_{t+1}$11:        Update $Q(s_{t},a_{t})$ according to (Equation 20)12:        Next time step t⟵t+113:    **end while**14:**end for**

## 5. Implementation

Using the above scenario, we conduct an evaluation by implementing an actual testbed that can generate and trade energy. In this testbed, we set two participants, the seller and buyer in the platform, and conduct several tests for the case in which an actual energy transaction could be made. The details of the tests are as follows: (1) Check the possibility that it is possible to transfer the values to the stepper motor, which is obtained through reinforcement learning. (2) Set the environment in which the stepper motor can operate the solar panel module. (3) Evaluate whether energy is transferred between two participants. (4) Check that the transmission is done, and ETHs are exchanged during the transaction process. Furthermore, we implement the proposed system as web pages and mobile applications so that users could perform transactions smoothly and obtain the necessary information they needed.

### 5.1. Testbed Hardware

The proposed testbed uses an ‘SCM 5WA’ solar cell module (solar panel), ‘ESC 1206’ charge controller, and ‘KB 4.5 Ah 12 V’ battery. The 42-angle stepper motor ‘NEMA17’ is used to move the solar panel. Additionally, the ‘L298N 1 channel’ is used to connect the motor to Raspberry Pi. At this time, the ‘DC-DC BUCK/BOOST’ converter is used to adjust the voltage because the voltage sizes are all different.

It moves the stepper motor and controls the solar cell module through the action values obtained through reinforcement learning. The controller stores the incoming energy in the battery and manages the energy stored in the battery for use. Using this function of the controller, the testbed checks that when the seller sends the specified amount of energy to the buyer, all that energy is sent to the buyer. Additionally, it connected the LED light bulb to the controller of the buyer to check whether the energy received from the seller can be used immediately. Figure 5 shows the blueprint of our energy transaction. Furthermore, Figure 6 is a real testbed that we implemented.

### 5.2. Software

The proposed testbed uses Solidity to write the terms of the smart contract as represented in Algorithm 2 and proceeds with the transaction.

To proceed with Ethereum’s deal, the user needs a personal wallet, and the user uses MetaMask [25], Google’s extension program that allows users to safely manage Ethereum’s personal wallet. MetaMask can only be used with Chrome, a PC browser; therefore, the proposed testbed creates a web page for energy trading to enable smooth trading. Energy generation is simulated using Python 3.7. We implement Algorithm 1 using Python and control the action value obtained from it.

A web page, which provides the real deal experience implemented in Figure 7a, gives the user the real-time price of Ethereum, the number of Ethereum and token users, and the address of the user’s personal wallet. In Figure 7b, a compartment enters the amount of energy a user wants to purchase. If the order quantity is entered, the price will be automatically converted. When the user presses the “Purchase of Energy” button, a pop-up window appears informing them that the energy transaction will be performed at Figure 7c. When the deal is processed, it appears that MetaMask is executed as shown in Figure 7d and Ethereum will be sent to the seller’s wallet. Upon the approval of the transaction, it may be confirmed that transaction approval is conducted as shown in Figure 7e,f and then completed. Following this process, the number of Ethereum and tokens currently in use are updated in real time.
**Algorithm 2 **Transaction Condition in Smart Contract.1:**if** Seller’s energy is bigger than Buyer’s requirement **then**2:    build smart contract3:    **while** Exist Energy **do**4:        **while** Exist token **do**5:           **if** Check balance of token **then**6:               **if** Check balance of Ether **then**7:                   swap token and Ether8:                   record transactions on the blockchain9:               **end if**10:           **end if**11:        **end while**12:    **end while**13:**end if**

In the case of the proposed testbed, we implement an Android mobile application using Android Studio, Java, and Etherscan for the convenience of market participants. However, because MetaMask’s characteristics make transactions impossible on mobile devices, the application mainly serves to provide information to the user. Figure 8a is the main page of the mobile application showing the current Ethereum’s market price. Additionally, Figure 8b provides contract address information and does not provide the user’s account address because MetaMask cannot be installed on a mobile device. In Figure 8c, the price can be verified by entering the desired quantity of purchases. Here, Figure 8d shows a graph of price and volume. Finally, by entering the user account address information shown in Figure 8e, we can check the user’s transaction history as depicted in Figure 8f.

After conducting actual energy transaction simulations using the hardware testbed and web page, we could check that the energy is moved. Additionally, the LED light bulb is checked through the energy received and it is proved that the energy received could be used, and the Ethereum exchanged through the transaction is reflected and recorded in real time.

## 6. Evaluation

To analyze the performance of the proposed system, we conduct a simulation by adjusting various parameter variables. Furthermore, the proposed system is compared with other algorithms to determine whether it is effective. To proceed with the simulation, we generate solar energy from several buildings at KAIST over a long time and used this value as the actual training data.

Figure 9 is the result of ten tests each with various parameter values, averaging the amount of solar energy that can be generated per day as a result. In Figure 9a, we see that 8000 episodes are needed to achieve optimal learning effects. Additionally, Figure 9b shows that the learning rate α is the best parameter value of 0.1. However, Figure 9c shows that the discount factor $\gamma_{i}$ does not have a significant impact on the training result.

Several algorithms are compared to analyze the performance of the proposed system. (1) A static panel that generates energy without moving the panel. (2) Regular panels move at the same angle every time and generate energy. (3) The heuristic panel moves in such a way that it can generate the most energy during that period. Finally, (4) the proposed system moves according to the completed Q-table and generates energy.

The simulation results compared to the various systems are shown in Figure 10. The proposed system generates 35% and 16% more energy than the fixed and regular panels, respectively. Additionally, the heuristic algorithm, which moves to the optimal location, collects 0.06% more energy than the proposed system, indicating that there is no significant difference in energy collection.

However, the proposed system no longer needs to train once the Q-table converges. Therefore, we believe that the proposed system will be more effective than a heuristic algorithm that should continue checking the energy collection. As a result, we have confirmed that the proposed system is effective in terms of solar panel generation compared with the other operation methods by comparing the energy generation rates and their effectiveness.

To analyze the profits of each market participant according to the auction progress, we pre-determine the value of variables and progress the numerical simulation results. For the sales, electricity rate $C_{sell}$ and future expected cost $C_{op}$ are set to KRW 0.1 and KRW 0.5. In addition, battery efficiency η is set to 0.9, and the amount of energy generated by each time period is set to be located between 1–10 kWh. In the case of the buyer, the price of electricity purchased through the conventional market is set to 0.08–0.24 per kWh considering the progressive billing system in the Republic of Korea. In addition, we set the average amount of electricity required for each time period to 5 kWh for the target of the small-scale power consumer. At this time, numerical simulation results are progressed to analyze the strategies of each market participant.

As depicted in Equation (Equation 5), a value of battery efficiency is related with the seller’s decision. Here, low efficiency means a large energy loss in storing progress, so sellers take a strategy to sell energy rather than store it. According to Figure 11a, when the battery efficiency is less than 0.2, it can be confirmed that the seller is taking a strategy to sell the whole energy as the financial profit when storing progress is relatively low. In addition, the seller’s strategy according to the relative electricity rates is shown in Figure 11b. Here, an increase in the value of the auxiliary variable α means that the current energy sale price $C_{sell}$ increases or the future expected price $C_{op}$ decreases. In this case, it can be seen that the seller takes a strategy to sell more energy at the present time and try to reduce the energy stored.

For the case of the buyer, their strategies are related with the amount of transaction energy, which is determined by the seller. When the transaction energy *E* is greater than a buyer *i*’s required energy $D_{i}$, the buyer pays an unnecessary cost as mentioned in Equation (Equation 6). To prevent such a problem, the buyer tries to reduce the bidding price as the amount of energy in the auction increases, as depicted in Figure 12a. Here, the buyer’s bidding price is closely related to the amount of energy in the auction, but also closely related to the success rate according to the bidding price. The fact that the higher value of the auxiliary variable $\gamma_{i}$ means that the probability of successful bidding is increased even at the bidding price of the buyer *i* is low. Therefore, when $\gamma_{i}$ increases, the bidding price tends to decrease as depicted in Figure 12b. In addition, buyer’s bidding price is also affected by the electricity price as argued in Equations (Equation 10) and (Equation 12). From these equations, it is possible to estimate that the buyer’s bidding price tends to increase proportionally according to the increment of market price.

Furthermore, we verify that energy trading is going smoothly with our testbed. On the implemented platform, when the buyer purchases energy from the seller, the energy moves from the seller to the buyer, as shown in Figure 13. To check whether the energy moved, we mount an LED sensor on the buyer’s testbed. We confirm that energy trading occurs by checking that the LED sensor is lit up as much as the energy received when the buyer receives the energy.

## 7. Conclusions

This study proposes a solar energy generation and transaction platform (EggBlock) using blockchain technique. The proposed EggBlock platform can maximize solar energy generation by controlling the angle of generator, and balance the power supplement by utilizing the proposed energy transaction scheme. Through the various simulation results, we showed that the dynamic panel control mechanism on the EggBlock converges well to the optimal directions with short iterations, which results in an average energy generation gain of 35%. In addition, in the energy trading, we also analyzed the strategies of each market participant according to the changes in the electricity price and environment of the actual power system. In our future study, the uncertainty of solar energy generation should be considered both in the energy trading and generation to address the practical concerns.

## Figures and Tables

**Figure 1 sensors-22-02410-f001:**
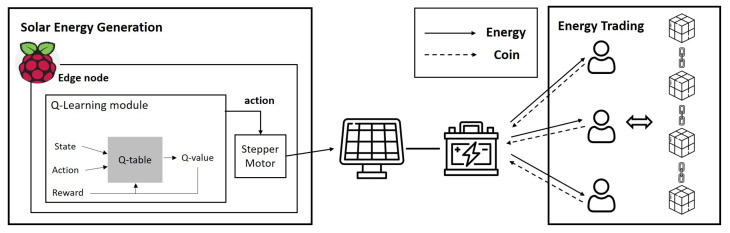
System model.

**Figure 2 sensors-22-02410-f002:**
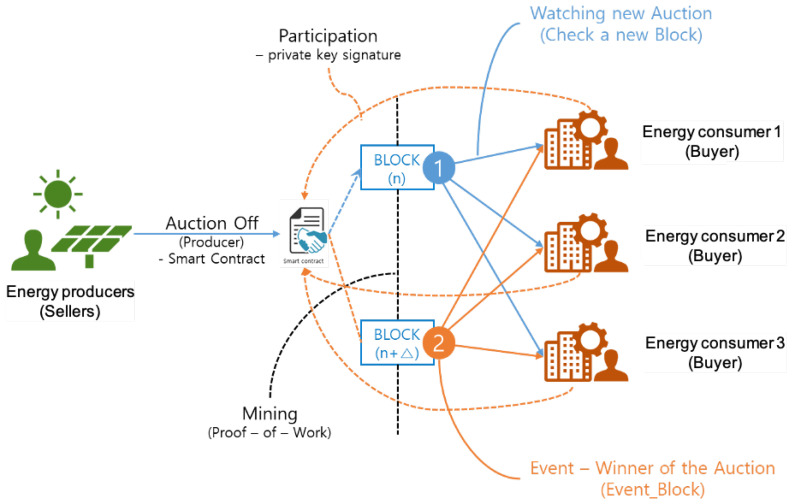
Auction-based energy transaction model.

**Figure 3 sensors-22-02410-f003:**
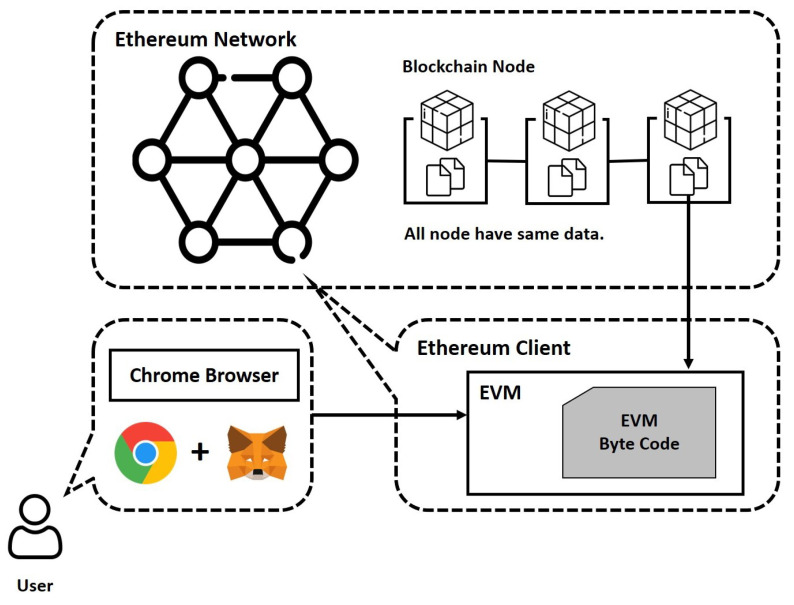
Ethereum architecture.

**Figure 4 sensors-22-02410-f004:**
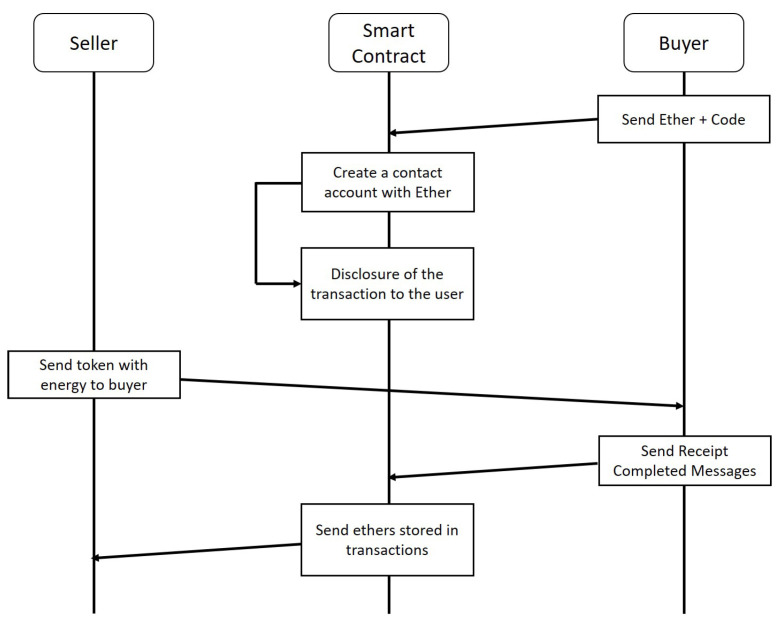
Sequence diagram for the energy trading.

**Figure 5 sensors-22-02410-f005:**
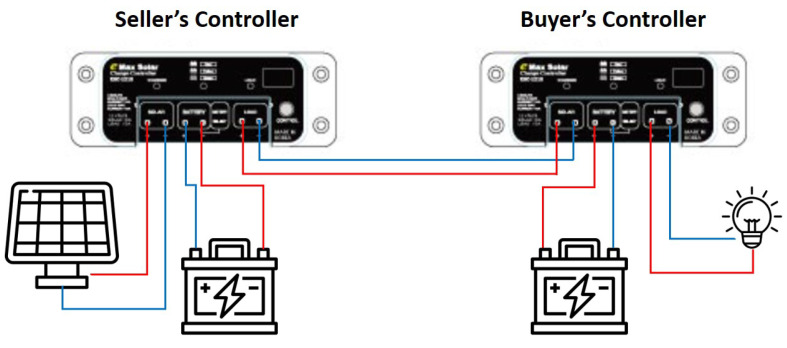
Blueprint of controllers in testbed.

**Figure 6 sensors-22-02410-f006:**
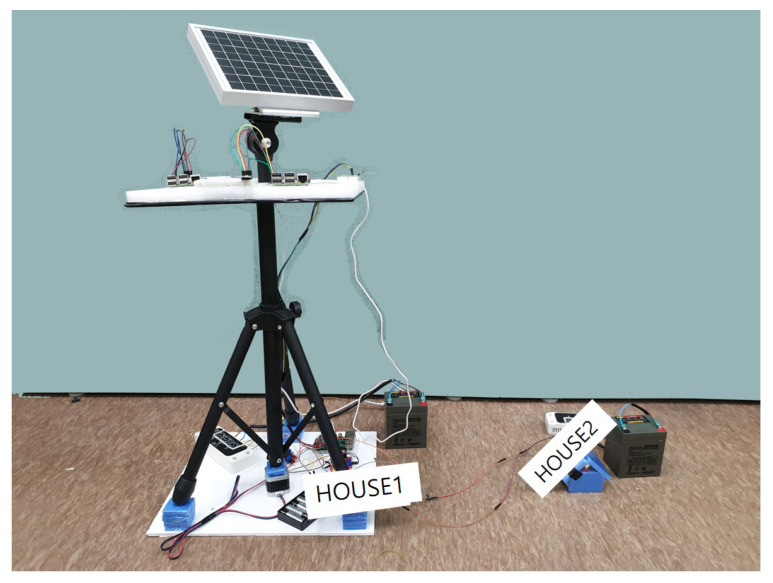
Testbed of platform.

**Figure 7 sensors-22-02410-f007:**
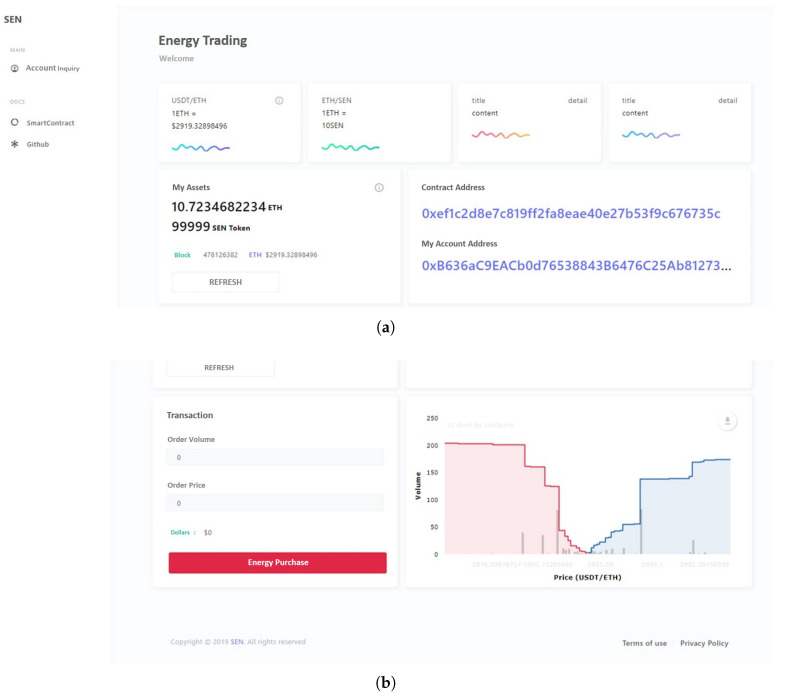
Process of energy trading in web page. (**a**) Real time user’s information; (**b**) Purchase section; (**c**) Purchase progress pop-up window; (**d**) Ethereum transaction preview; (**e**) Transaction in progress; (**f**) Transaction completion.

**Figure 8 sensors-22-02410-f008:**
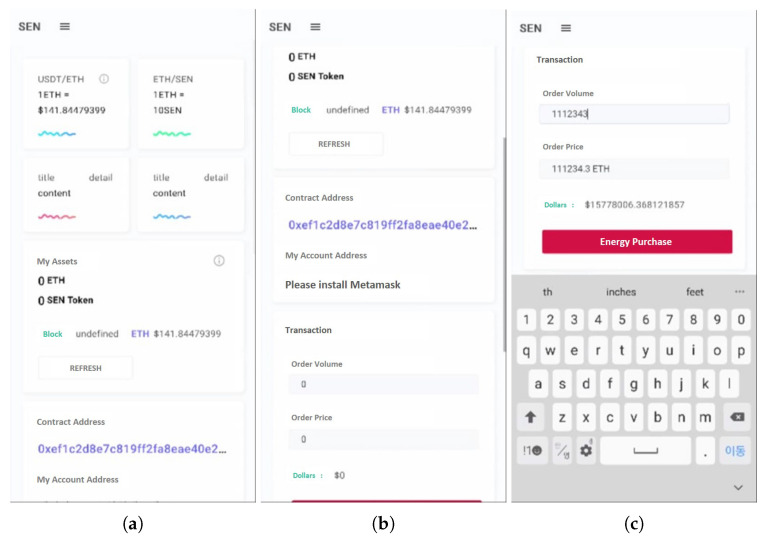
Provided information in mobile application. (**a**) Main page; (**b**) Contract address; (**c**) Market price; (**d**) Price with volume; (**e**) Account lookup; (**f**) Transaction history.

**Figure 9 sensors-22-02410-f009:**
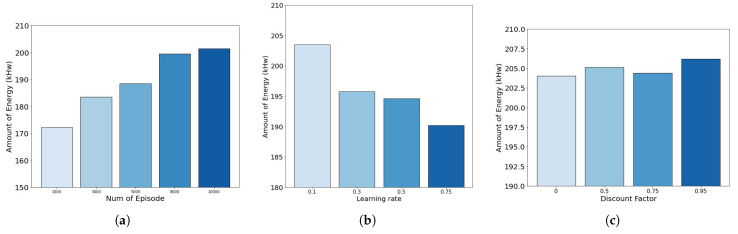
Adjusting parameters according to the amount of solar energy generated per day. (**a**) Adjusting number of episode; (**b**) Adjusting learning rate; (**c**) Adjusting discount factor.

**Figure 10 sensors-22-02410-f010:**
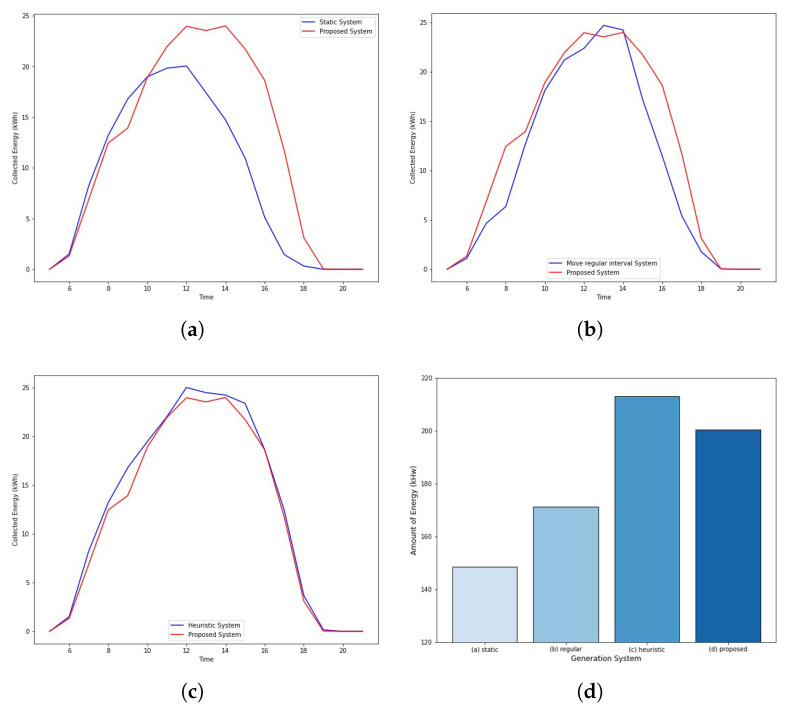
Simulation result. (**a**) Compared with static system; (**b**) Compared with regular system; (**c**) Compared with heuristic system; (**d**) Total amount of generated energy.

**Figure 11 sensors-22-02410-f011:**
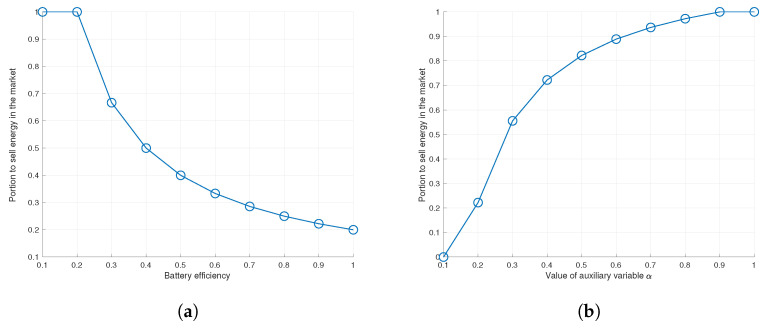
Determination of transaction ratio of seller according to environmental changes. (**a**) Compare with static system; (**b**) Compare with static system.

**Figure 12 sensors-22-02410-f012:**
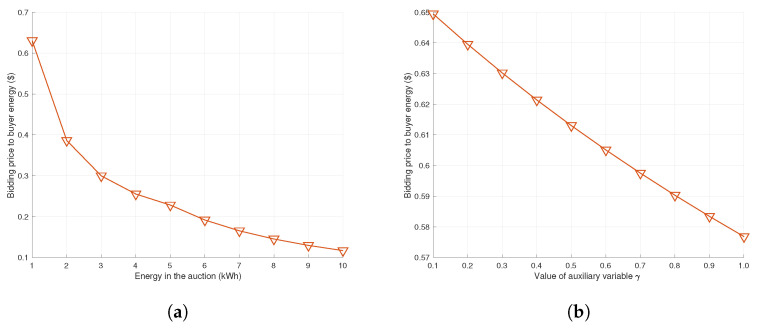
Determination of bidding cost of buyer *i* according to environmental changes. (**a**) Compared with static system; (**b**) Compared with static system.

**Figure 13 sensors-22-02410-f013:**
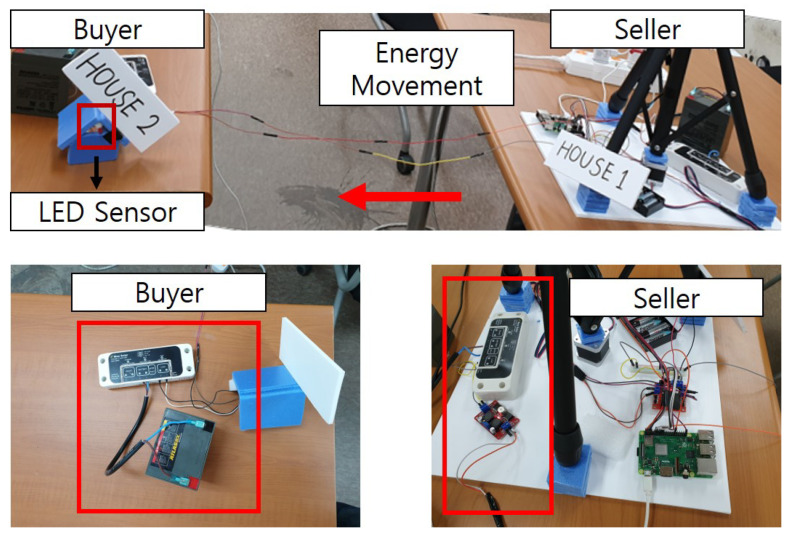
Energy trading test.

## Data Availability

Not applicable.

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
