# Peer review of "EggBlock: Design and Implementation of Solar Energy Generation and Trading Platform in Edge-Based IoT Systems with Blockchain"

_sensors, 2022, doi:10.3390/s22062410_

Round 1

Reviewer 1 Report

This paper is of interest and presented well. To reach publication level, please address the following concerns:

  1. In the abstract, the methodology of the suggested approach should be revised. There are some extra explanations. In my view, the authors can describe their methodology in a more informative way.
  2. The contributions of the paper are stated in six bullet points. It seems that some points are not novelties/contributions. The authors are recommended to shorten the contributions and point out the key novelties.
  3. To judge the contributions, a mature literature review is required. Please survey more papers in the section of "related works". Barely any studies are reviewed in recent years, i.e. 2021 and 2022. Please add more studies from prominent academic and industrial publishers.
  4. In the model of action-based energy transactions, what kinds of energy consumers are considered, e.g. type, scale? does it matter? or general energy buyers are modeled?
  5. Could you please increase the resolution of figure 7? in the current format, it is unclear.
  6. The same problem is for figure 10. It is very unclear. 
  7. Regarding the electricity price, have you investigated the impacts of electricity pricing on the problem? For example, in line 408, a constant price is offered. what about dynamic prices or time-of-use tariffs?
  8. Please extend the conclusion. Add some explanations to convey (1) brief motivation (2) informative methodology (3) key simulation results (4) main advantages over previous studies (5) limitations and challenges of the experimental study (6) suggestions for future research. Please revise this section accordingly.

Author Response

This submission is a revised version of our previously submitted paper. We thank the reviewers for their efforts and their detailed, helpful comments, which contributed to the improved quality of this paper.  We hope that you will find this new version suitable for publication. In the following, we provide detailed information on how we have modified the paper in accordance with the  reviewers’ comments. Furthermore, the reviewers’ comments appear in bold face below.

Best regards,
Authors of the paper, for all reviewers

Reviewer 2 Report

This paper proposed a blockchain-based energy trading platform for solar energy trading. Overall, this paper is well-prepared with some comments to be considered as follows.

  • Expand section 2 to include more blockchain technology in general and the difference in various blockchain technologies used in energy applications. The benefits of blockchain technology need to be highlighted versus the conventional methods.
  • Line 231 and 232. The energy transaction is carried out under the blockchain technique. Please elaborate more on the specific blockchain technique.
  • Section 3.1 explains adjusting the solar panel position may not ensure that a specific amount of electricity is generated. How can a certain amount of power be guaranteed in energy generation? It may be beneficial to address some of the uncertainties in renewable energy generation in the evaluation section.

Minor

  • Capitalized section 3.2
  • Check figures font size, and some are way too small; for example, Figure 9-10

Author Response

(The authors gave the same response as above.)

Round 2

Reviewer 1 Report

The authors have addressed my concerns.